# Heterogeneous Impact of Economic Policy Uncertainty on Provincial Environmental Pollution Emissions in China

**Wei Yang \*, Yifu Zhang and Yuan Hu**

School of Economics and Management, Shanghai Ocean University, Shanghai 201306, China;
m190501057@st.shou.edu.cn (Y.Z.); zhaodan2021@yeah.net (Y.H.)
**\*** Correspondence: m200501185@st.shou.edu.cn

**Abstract:** With China's proposal of carbon peak and carbon neutral goals, its trend of economic development has shifted from pursuing high-speed economic development to high-quality development. However, for the past few years, with the increasing global economic policy uncertainty, fluctuations in the world economy, especially emergent through public events such as COVID-19, affect investment and consumption, and thus indirectly affect the realization of the dual carbon target. Economic policy uncertainty plays an increasingly important role in many factors affecting environmental pollution. We conducted an empirical test on sample data, which are from 30 provinces and autonomous regions in China from 2008 to 2020, to further study the impact of economic policy uncertainty on environmental pollution emissions. We found that: (1) Economic policy uncertainty is inversely related to the emission of environmental pollution, and the consumption effect brought by economic policy uncertainty is more than the investment effect. This means that, with the economic policy uncertainty index increasing, the comprehensive index of environmental pollution emissions is lower, and the environmental pollution emissions are lower; (2) Compared with provinces with an average level of economic development, the impact of economic policy uncertainty on environmental emissions is deeper in developed provinces.

**Keywords:** economic policy uncertainty; comprehensive index of environmental pollution discharge; night light data; investment effect; effect of consumption

## 1. Introduction

Situated against the background of economic globalization, environmental problems caused by pollution emissions, especially greenhouse gas emissions, have become one of the biggest, widest, and most far-reaching challenges faced by human beings so far. From the Kyoto Protocol to the Paris Agreement, countries around the world are constantly seeking a balance between economic development and environmental pollution. At present, the world is pumping 51 billion tons of greenhouse gases into the atmosphere every year, and this figure will remain high until countries actually reduce their emissions. Against this backdrop, the parties at the United Nations Framework Convention on Climate Change agreed to set a timetable for carbon neutrality once emissions in most developed countries peak. As the world's largest developing country and major energy consumer, China has set a target. It is called the "3060 target", and in 2021, it was included in the government work report for the first time. On 21 September 2021, in his speech at the General Debate of the 76th Session of the United Nations General Assembly, President Xi declared: "China aims to achieve carbon peak by 2030 and carbon neutral by 2060. China will strongly support the green and low-carbon energy development of developing countries and will not build overseas coal power projects".

However, when reviewing the global economic development environment, many crucial factors still have an evident impact, including the global financial crisis in 2008, Brexit, the US election, the Us–China Trade Dispute in 2019, and the COVID-19 epidemic, which

has affected the whole world since 2020 and continues to spread. Such unpredictable global black swan events with great suddenness and extreme impacts aggravate the uncertainty of the global economy, increasing the difficulty of international policy coordination, increasing geopolitical risks, and increasing economic instability factors. The sudden increase in economic policy uncertainty (EPU) caused by such black swan events will bring severe damage to the realization of the set goals of countries around the world. The global average EPU index in 2020 has reached 328.4, an increase of 1.6 times compared to the period of the financial crisis. In addition, EPU in major countries around the world has also increased several times.

When economic policy uncertainty gradually increases, sustainable development becomes more urgent; therefore, the relationship between economic policy uncertainty and environmental pollution emissions has been deeply discussed by scholars. Pirgaip et al. [1] found a close bidirectional relationship between EPU and carbon emissions, which is based on the data of carbon emissions and EPU of G7 countries from 1998 to 2018. However, the two-way relationship between EPU and environmental pollution emissions has not formed a unified conclusion in the academic circle. There are three main opinions: one opinion believes that there is a positive correlation between economic policy uncertainty and carbon dioxide emissions. Wang et al. [2] found that economic policy uncertainty is positively correlated with $CO_2$ emissions in the long term. Jiang et al. [3], based on the data of US sectors, found that except for the commercial sector, the US EPU had a significant Granger causality, which means that the tail of the growth of carbon emissions is distributed in the power sector, industrial sector, transportation sector, and housing sector. Adams et al. [4] used the World Uncertainty Index (WUI), which is another measure of uncertainty alongside the EPU, to study 10 countries in the world, and the result shows that the increase in World Uncertainty Index will result in the increase in carbon dioxide emissions; the result held true in both the short and long term. Another opinion takes the opposite view. Chen et al. [5] performed an empirical analysis, based on the panel data from 15 countries all over the world; the empirical analysis showed that the bigger the EPU index, the fewer the carbon dioxide emissions per capita. Danish et al. [6] agreed with this. The third opinion argues that the relationship is related to the period. Adedoyin et al. [7] found that the increase in EPU will reduce $CO_2$ emissions in the short run, and will lead to an increase in $CO_2$ emissions in the long run. Most scholars study the economic policy uncertainty and the environment pollution, especially the relationship between carbon dioxide emissions; however, few studies have examined the impact of the economic policy uncertainty on the environmental pollution discharge mechanism (and rarely in a country like China), with energy consumption as the research object, and with a continued discussion on the heterogeneity between the provinces in China.

We illustrate this transmission mechanism from the perspective of the volatility between economic policy uncertainty and other markets. The current literature research shows that the changes in EPU are closely related to the energy price market and stock market price.

Some scholars believe that the fluctuation of international oil price is the main reason for the fluctuation of economic policy uncertainty. Rehman [8] found that economic policy uncertainty in India, Japan, Spain, and other countries was closely related to the international oil price level. Further, Degiannakis et al. [9] believe that fluctuations in international oil prices may lead to changes in the future cash flow of relevant oil companies, which in turn will lead to fluctuations in the company's stock prices and ultimately affect the uncertainty of economic policies through the stock market. Li et al. [10] investigated the volatility spillover effect between oil price and EPU before and after the financial crisis, and the empirical analysis showed that there was no volatility spillover effect between oil price and EPU before and after the financial crisis on the premise of excluding the impact of exogenous shocks. However, exogenous shocks can enhance the transmission of volatility between oil price and EPU, taking the financial crisis as the node. Kang et al. [11] found through empirical analysis that the positive oil price shocks caused by higher precautionary

oil demand are significantly correlated with increased uncertainty in US economic policy. Ji et al. [12] discussed the impact of EPU on energy prices and found that there is usually a negative correlation between changes in EPU and energy returns. Lin et al. [13] highlight that the EPU of importing countries is more susceptible to the impact of oil price fluctuations. A higher EPU index will lead to changes in international oil supply and demand, which will also affect energy price changes, thus affecting energy consumption and, ultimately, environmental pollution emissions. Danish et al. [6] concluded that the higher the EPU index, the greater the energy consumption, and the greater the environmental pollution emissions. Pirgaip et al. [1], through analyzing the EPU of G7 countries and the influence of the relationship between energy consumption and $CO_2$ emissions, found that the Japanese EPU increased energy consumption, while in the USA and Germany, an increase in EPU index increased $CO_2$ emissions.

Secondly, the fluctuation of the stock market price is closely related to the fluctuation of EPU. Baker et al. [14] tested the enterprise micro-subject data and found that the increase in EPU index led to the decrease in investment level, especially in industries which are very sensitive to policy changes, such as the infrastructure construction industry. Brogaard et al. [15] found that every 1% increase in the EPU index would increase the 3-month expected return rate by 1.5%. Pastor et al. [16] constructed a general equilibrium model to study the relationship between EPU index and stock price. They found that the fluctuation of the EPU index often accompanied the fluctuation of the stock market, which was especially obvious in the economic recession. Liu et al. [17] used the heterogeneous autoregressive model and combined it with the daily EPU index to study its prediction effect on stock market volatility, and found that the uncertainty index could accurately predict stock market volatility. Arouri et al. [18] found a significant negative correlation between the EPU index and the stock return rate through panel data regression, and discovered that the impact of EPU on the stock return rate will last for a long time.

After discussing the reasons for the fluctuation of EPU, we have a deeper understanding of the influence mechanism of EPU on environmental pollution emissions. Wang et al. [2] proposed that the EPU has two main impacts on environmental pollution emissions, namely the "investment effect" and the "consumption effect". On the one hand, the frequent changes of economic policies of various countries will increase the investment cost and risk of enterprises, and will then inhibit the investment of enterprises until the uncertainty disappears. On the other hand, Bloom et al. [19] believed that investors would delay investment due to uncertainty about the future market and uncertainty about the effect of policies. Farzin et al. [20] found through research that the increase in the EPU index would bring the reduction of enterprises' investment in cleaning equipment and technology, which is one of the important ways for enterprises to improve green productivity. Another way for enterprises to improve green productivity and reduce pollutant emissions is to adopt green energy and renewable energy. However, the increase in uncertainty of economic policies will lead enterprises to adopt a wait-and-see attitude, reduce the use of clean energy, continue to use traditional energy, and increase environmental pollution emissions. Therefore, the investment effect brought about by the EPU will increase the emission of environmental pollutants.

The consumption effect and the investment effect have the opposite impact. Claeys [21] estimated the spillover effects of the uncertainty shocks from the quarterly data of developed countries and emerging economies from the first quarter of 1990 to the third quarter of 2014, using the panel VAR method, and found that the investment and consumption in emerging markets suffered a sharp decline with the global spread of uncertainty. Mumtaz et al. [22] used quarterly data from 1970 to 2015 in the USA to show that the adverse impact of government debt uncertainty on output, consumption, consumer confidence, investment, and business confidence was great and long-lasting. Coibion et al. [23] conducted a survey of 10,000 eurozone countries and found that the increase in the EPU index in 2020 would lead to a decrease in household expenditure. Therefore, the "consumption effect" is when the EPU index increases, the initiative of social production will

be reduced. For residents, the consumption level of individual residents will be reduced, and the reduction of demand for consumer goods will adversely affect the whole society to reduce social supplies. For enterprises, it will reduce the input of the socialized production of enterprises, and it will then reduce the amount of environmental pollution emissions from investment and consumption.

Similarly, in terms of analyzing the impact of heterogeneity brought on by different levels of economic development, most scholars distinguish between total GDP, per capita GDP, or disposable income, while Wu [24] believes that the authenticity and accuracy of GDP data of such differentiated regions need to be discussed. Adding GDP data to model estimates is bound to affect the results. It is worth noting that Ghost et al. [25] used remote sensing data of night light to depict the intensity of surface economic vitality, and Chand et al. [26] also believed that the intensity of night light brightness was the embodiment of regional economic development level. Chen et al. [27] and Henderson et al. [28] also found that satellite observation data can accurately reveal the local economic development level through the empirical analysis.

Through the review of the relevant literature, it can be seen that the existing literature rarely studies the heterogeneity within the same economy, let alone of China, a country with a large population and resources, which contributes significantly to the sustainable development of the world. Similarly, most scholars have studied the relationship between EPU and environmental pollution emissions, especially carbon dioxide emissions, but few have studied the mechanism of the impact of the EPU index on environmental pollution emissions. Therefore, it is of great significance to study the impact and mechanism of the EPU index on environmental pollution emissions in China, and to provide suggestions for global sustainable development. At the same time, we use nighttime light data to measure the differences of economic development levels among provinces in China, so as to study whether the economic development level is an important factor affecting environmental pollution emissions due to economic policy uncertainty, which is a novel method when compared with previous literature studies. We try to make up for the deficiencies of the existing literature and enrich the relevant research in the above two aspects. The comprehensive index of environmental pollution emissions constructed in this paper is calculated based on the total amount of waste water discharged, the amount of sulfur dioxide in waste gas, and the production of general industrial solid waste, to measure the level of environmental pollution.

Therefore, based on the above literature review and analysis, the following two hypotheses are proposed by us:

**Hypothesis 1 (H1).** *The investment effect of EPU on environmental pollution emissions is less than the consumption effect; that is, the higher the EPU index, the lower the comprehensive index of environmental pollution emission, and the lower the environmental pollution emissions.*

**Hypothesis 2 (H2).** *Compared with provinces with an average level of economic development, the impact of economic policy uncertainty on environmental emissions is deeper in developed provinces.*

## 2. Data Selection and Model Selection

### 2.1. Variable Selection and Data Sources

As the explained variable, we intend to use the comprehensive pollution emission index of each province. As the core explanatory variable, we use the EPU of China to study the impact of economic policy uncertainty on the comprehensive pollution emission index of each province. In order to study the heterogeneity of the provinces in China, we use the 30 provinces in China as research objects, including data from 2008 to 2020. Because data from Tibet for certain years was difficult to obtain, it was excluded. The comprehensive carbon emission index of each province was used to describe the pollution emission of each province in China, the index of EPU of China was used to describe the uncertainty of each province's economic policy, the nighttime light data

was used to describe the development of quality, and we measured the level of foreign investment by using the total amount of foreign capital actually utilized. The calculation of the comprehensive index of provincial environmental pollution emission comes from the index algorithm of Wang [29] and Zhou et al. [30], using the principal component analysis method and the entropy weight method, respectively. EPU data were from www.policyuncertainty.com/, accessed on 11 March 2022. Night light raw data were from www.eogdata.mines.edu/products/vnl/, accessed on 11 March 2022. However, the Defense Meteorological Satellite Program_Operational Linescan System V4 (DMSP_OLS V4) (2008–2011) and Visible Infrared Imaging Radiometer Suite_Day/Night Band V2 (VI-IRS_DNB V2) (2012–2020) are essentially different and incompatible. In this paper, annual DMSP data were corrected based on the results of Wu et al. [31], Zheng et al. [32], and Ma et al. [33]. Then, the annual VIIRS data were de-noised, and the two sets of overlapping years, 2012 and 2013, were extracted for sensitivity analysis, and the optimal fitting parameters were selected. Then, according to the selected optimal parameters, the annual data of VIIRS (2012–2020) were calculated into the data of DMSP (2012–2020). Finally, the synthetic VIIRS (1992–2020) data set was constructed. The total amount of foreign direct investment (FDI) actually utilized came from the wind database, provincial statistical yearbook, provincial statistical bulletin, and the official website of the provincial statistics bureau.

Table 1 shows the two core variables of the model in our research, namely, the comprehensive pollution emission index of each province (CE) and the EPU. The CE constructed in this paper adopts two methods. The comprehensive pollution emission index of each province calculated by entropy weight method is the data required for the robustness test in this paper. First of all, in terms of the comprehensive pollution emission index, the data obtained by principal component analysis method show that the top three provinces are Shandong, Guangdong, and Hebei, with an average of 1.823, 1.70, and 1.442, respectively. However, Hainan, Tianjin, and Beijing have the lowest environmental pollution composite index, which are −1.425, −1.168, and −1.087, respectively. The comprehensive pollution emission index of each province calculated by entropy weight method shows that the top three provinces are Hebei, Shandong, and Shanxi, with average values of 0.663, 0.648, and 0.565, respectively. Hainan, Tianjin, and Beijing remained the lowest among all provinces, with indexes of 0.02, 0.067, and 0.073, respectively. Secondly, in terms of EPU, the average value of China's EPU from 2008 to 2020 is 319.245.

**Table 1.** Descriptive statistical analysis of core variables.

| Variables | CE (Principal Component Analysis) | | | CE (Entropy Weight Method) | | | EPU | | |
|---|---|---|---|---|---|---|---|---|---|
| Provinces | Mean | Max | Min | Mean | Max | Min | Mean | Max | Min |
| Beijing | −1.087 | −0.979 | −1.226 | 0.073 | 0.116 | 0.045 | | | |
| Tianjin | −1.168 | −1.065 | −1.301 | 0.067 | 0.095 | 0.037 | | | |
| Hebei | 1.442 | 1.862 | 0.612 | 0.663 | 0.728 | 0.486 | | | |
| Shanxi | 0.918 | 1.299 | 0.676 | 0.565 | 0.603 | 0.477 | | | |
| Inner Mongolia | 0.802 | 1.599 | 0.517 | 0.518 | 0.636 | 0.429 | | | |
| Liaoning | 0.890 | 1.172 | 0.547 | 0.523 | 0.635 | 0.431 | | | |
| Jilin | −0.727 | −0.700 | −0.785 | 0.158 | 0.182 | 0.141 | | | |
| Heilongjiang | −0.449 | −0.131 | −0.588 | 0.214 | 0.257 | 0.184 | | | |
| Shanghai | −0.682 | −0.365 | −0.927 | 0.143 | 0.235 | 0.097 | 319.345 | 791.470 | 98.890 |
| Jiangsu | 1.408 | 1.678 | 0.808 | 0.526 | 0.599 | 0.435 | | | |
| Zhejiang | 0.283 | 0.575 | −0.094 | 0.307 | 0.378 | 0.256 | | | |
| Anhui | 0.009 | 0.142 | −0.180 | 0.311 | 0.349 | 0.277 | | | |
| Fujian | −0.179 | 0.103 | −0.400 | 0.250 | 0.307 | 0.194 | | | |
| Jiangxi | −0.032 | 0.354 | −0.225 | 0.306 | 0.382 | 0.257 | | | |
| Shandong | 1.823 | 2.200 | 1.253 | 0.648 | 0.709 | 0.570 | | | |
| Henan | 0.996 | 1.340 | 0.399 | 0.479 | 0.580 | 0.386 | | | |
| Hubei | 0.073 | 0.149 | −0.040 | 0.299 | 0.341 | 0.268 | | | |

**Table 1.** *Cont.*

| Variables | CE (Principal Component Analysis) | | | CE (Entropy Weight Method) | | | EPU | | |
|---|---|---|---|---|---|---|---|---|---|
| Provinces | Mean | Max | Min | Mean | Max | Min | Mean | Max | Min |
| Hunan | 0.176 | 0.318 | 0.000 | 0.311 | 0.371 | 0.271 | | | |
| Guangdong | 1.701 | 2.045 | 1.134 | 0.533 | 0.641 | 0.470 | | | |
| Guangxi | −0.109 | 0.703 | −0.456 | 0.268 | 0.432 | 0.205 | | | |
| Hainan | −1.425 | −1.314 | −1.562 | 0.020 | 0.026 | 0.017 | | | |
| Chongqing | −0.535 | −0.295 | −0.656 | 0.176 | 0.241 | 0.142 | | | |
| Sichuan | 0.607 | 0.925 | 0.418 | 0.413 | 0.489 | 0.350 | | | |
| Guizhou | 0.026 | 0.867 | −0.285 | 0.309 | 0.444 | 0.244 | | | |
| Yunnan | 0.026 | 0.634 | −0.510 | 0.334 | 0.440 | 0.238 | | | |
| Shaanxi | −0.183 | −0.037 | −0.265 | 0.267 | 0.303 | 0.229 | | | |
| Gansu | −0.745 | −0.583 | −0.906 | 0.159 | 0.176 | 0.141 | | | |
| Qinghai | −1.027 | −0.650 | −1.445 | 0.138 | 0.206 | 0.053 | | | |
| Ningxia | −0.978 | −0.749 | −1.206 | 0.114 | 0.162 | 0.092 | | | |
| Xinjiang | −0.318 | 0.142 | −0.735 | 0.244 | 0.356 | 0.173 | | | |

Figure 1 shows the nighttime light situation in China in 2008, 2012, 2016, and 2020. During the development period of China from 2008 to 2020, the brightness of the night light in various provinces and autonomous regions has a trend of slowly expanding from North China to Yangtze River Delta, Pearl River Delta, and central China. Nighttime lighting data are combined with the country's per capita GDP data, which are released by the National Bureau of Statistics of China. We will focus on the five provinces of Jiangsu, Zhejiang, Fujian, Shandong and Guangdong, because they are economically developed provinces in China, and because of the discussion around their economic development level; moreover, they have general heterogeneity in regards to the impact they experience of economic policy uncertainty on the environmental pollution emissions.

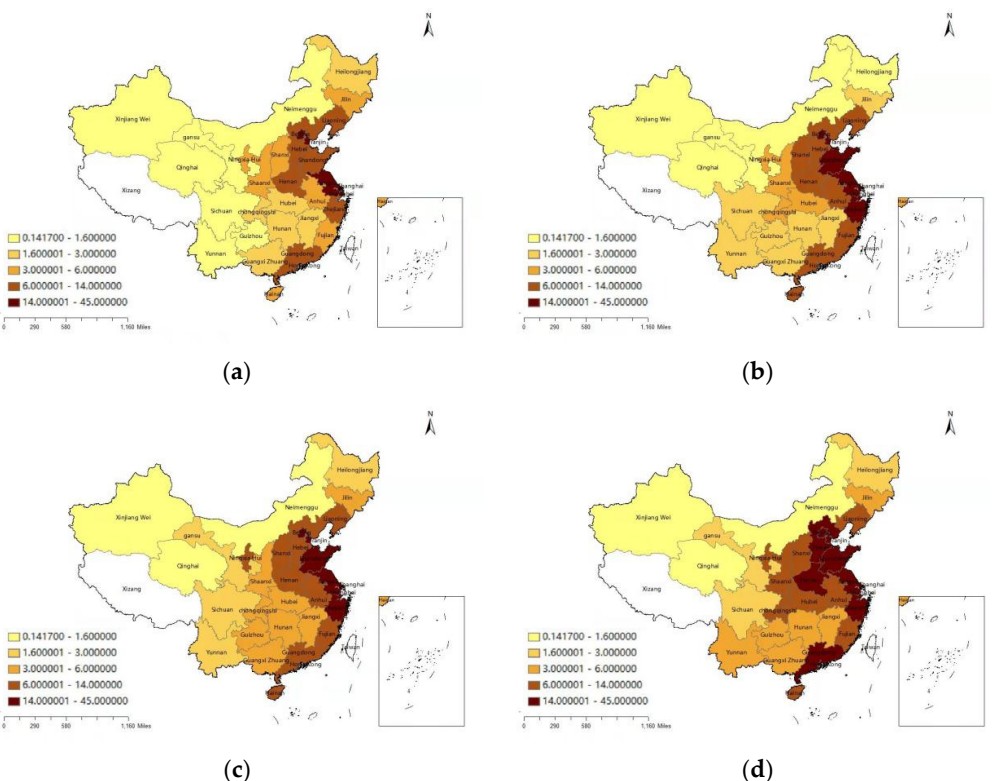

**Figure 1.** Night light data of Chinese provinces and autonomous regions in (**a**) 2008, (**b**) 2012, (**c**) 2016, and (**d**) 2020.

*2.2. Empirical Model*

The impacts of EPU on environmental pollution emissions of 30 provinces and autonomous regions in China are studied by taking the comprehensive pollution emission index (CE) of each province as the explained variable, and the economic policy uncertainty (EPU), total utilized foreign investment (FDI), and night light data (VIIRS) as the core explanatory variables. In order to distinguish the heterogeneous impact of EPU on environmental pollution emissions in developed provinces and ordinary provinces, a dummy variable was adopted to reflect the economic development level of each province, denoted as Deve. The economically developed provinces are denoted as 1 and the economically mediocre provinces as 0, and an interaction term between the provincial development level and the uncertainty of economic policy is determined, denoted as Deve*EPU. The model is as follows:

$$\text{CE}_{it} = \alpha_i + \beta_0 + \beta_{1i} * \text{FDI}_{it} + \beta_{2i} * \text{VIIRS}_{it} + \beta_{3i} * \text{EPU}_{it} + \beta_{4i}\text{Deve}_{it} * \text{EPU}_{it} + \mu_{it} \quad (1)$$

In view of the differences among different provinces, the panel variable coefficient model is constructed first, and the optimal model is selected according to the model estimation results. In addition, the individual fixed effect model or individual random effect model should be selected when constructing the model. The individual fixed effect model (FEM) is shown as follows:

$$\text{CE}_{it} = \alpha_i + \beta_0 + \beta_1 * \text{FDI}_{it} + \beta_2 * \text{VIIRS}_{it} + \beta_3 * \text{EPU}_{it} + \beta_4\text{Deve}_{it} * \text{EPU}_{it} + \mu_{it} \quad (2)$$

Although the intercepts between different study objects may vary, the intercepts of each study object do not change over time. The regression coefficient does not vary with the change of object or period. In this research, the random effects model (ECM) is shown as follows:

$$\text{CE}_{it} = \alpha_i + \beta_0 + \beta_1 * \text{FDI}_{it} + \beta_2 * \text{VIIRS}_{it} + \beta_3 * \text{EPU}_{it} + \beta_4\text{Deve}_{it} * \text{EPU}_{it} + \mu_{it}$$
$$\alpha_i = \alpha_1 + \varepsilon_i \quad (3)$$

## 3. Empirical Results of the Impact of EPU on Pollution Emissions

*3.1. Unit Root Test and Co-Integration Test*

We use data which are from 30 provinces in China to study the impact of EPU on environmental pollution emissions. In order to prevent the influence of pseudo-regression on the model, it is necessary to carry out a stability test on the panel data. Panel LLC, ADF, and Fisher PP tests are used to the conduct unit root test, and the results are shown below.

In Table 2, at the significance level of 10%, the original sequence of the variables of economic policy uncertainty (lnEPU), night light data (lnVIIRS), total actual utilized foreign capital (lnFDI), and environmental pollution emission efficiency (CE) did not pass all the tests under LLC, ADF, and Fisher PP tests. There is a unit root, which is a non-stationary sequence. After the first-order difference of each variable, the first-order difference sequence of each variable passes all the tests under the three tests, and there is no unit root, so it is a stationary sequence. Therefore, each variable is an integral sequence of first order.

The co-integration test is used to determine the purpose of the non-stationary sequence of whether there is a long-term equilibrium relationship between variables; it does this by the unit root test showing economic policy uncertainty (lnEPU), night lighting data (lnVIIRS), and the total amount of the actual use of foreign capital (lnFDI), as well as the environmental pollution emission efficiency (CE), which is a first-order single whole sequence, with the football association being the whole test conditions, and the panel Pedroni co-integration test being used for the co-integration test of the above variables. Table 3 shows the result.

**Table 2.** Unit root test of each variable.

| Variables | LLC Test Value | ADF Test Value | Fisher PP Test Value | Conclusion |
|---|---|---|---|---|
| lnEPU | 2.5869 (0.9952) | 7.4034 (1.0000) | 5.8223 (1.0000) | No stability |
| lnVIIRS | −3.7943 (0.0001) | 32.7924 (0.9984) | 27.3430 (0.9999) | No stability |
| lnFDI | −10.0644 *** (0.0000) | 129.5047 *** (0.0000) | 29.6681 (0.9996) | No stability |
| CE | −2.4696 *** (0.0068) | 63.2166 (0.3635) | 83.6318 ** (0.0236) | No stability |
| ΔlnEPU | −20.4110 *** (0.0000) | 398.4523 *** (0.0000) | 241.0424 *** (0.0000) | Stability |
| ΔlnVIIRS | −11.1694 (0.0000) | 217.8185 (0.0000) | 367.5354 (0.0000) | Stability |
| ΔlnFDI | −6.1521 *** (0.0000) | 125.2790 *** (0.0014) | 217.3059 *** (0.0000) | Stability |
| ΔCE | −3.7093 *** (0.0001) | 135.7208 *** (0.0000) | 526.3521 *** (0.0000) | Stability |

Note: Δ stands for first-order difference sequence; *** and ** represent significance at the significance level of 1% and 5%, respectively.

**Table 3.** Co-integration test results.

| | Statistic | *p*-Value |
|---|---|---|
| Modified Phillips–Perron t | 2.7811 | 0.0027 |
| Phillips–Perron t | −12.0940 | 0.0000 |
| Augmented Dickey–Fuller t | −11.4681 | 0.0000 |

The results of the co-integration test show that the corresponding *p*-values of the three test statistics are all less than 0.1, so the null hypothesis of "no co-integration relationship exists" is strongly rejected. Therefore, there is a long-term co-integration relationship between the variables.

### 3.2. Model Selection and Parameter Estimation

The F test and Hausman test were used to verify the model to determine the form of the model before establishing the panel model. In Table 4, the F test statistic value is 153.19 and the *p* value of the test statistic is 0.000, which rejects the null hypothesis of the mixed effect. The Hausman test value is 36.20, and the *p*-value of the test statistic is 0.000, which rejects the null hypothesis of the random effect. Therefore, we adopt the fixed-effect model.

**Table 4.** Model selection.

| | Statistic | *p*-Value |
|---|---|---|
| F test | 153.19 | 0.000 |
| Hausman test | 36.20 | 0.000 |

### 3.3. Analysis of Regression Results

In order to analyze the impact of EPU on environmental pollution emissions of provinces, through the unit root test and the co-integration test of variables, as well as the F test and the Hausman test of the model selection, this paper constructed a panel model for analysis, as shown below:

$$CE_{it} = \alpha_i + \beta_0 + \beta_{1i} * lnEPU_{it} + \beta_{2i} * lnVIIRS_{it} + \beta_{3i} * lnFDI_{it} + \beta_{4i}Deve_{it} * lnEPU_{it} + \mu_{it} \tag{4}$$

The specific model results are as follows. As can be seen from the Table 5 below, the coefficient of lnEPU is −0.105, which is significant at the significance level of 10%, indicating that the EPU has a significant negative impact on environmental pollution emissions. The

higher the EPU, the lower the comprehensive index of environmental pollution emissions. To study whether there is heterogeneity between the economically developed provinces, the coefficient of the interaction term introduced into the model shows that the negative impact of the EPU index on environmental pollution emissions in the economically developed provinces is 2.98 times that in the less economically developed provinces.

**Table 5.** Cross-sectional fixed effect regression results.

| Variables | Coefficient | Standard Error | T Statistic | *p*-Value |
|---|---|---|---|---|
| C | −0.90995 | 0.260322 | −3.50 | 0.000 |
| lnEPU | −0.10537 | 0.0573 | −1.84 | 0.066 |
| lnVIIRS | 0.315095 | 0.085816 | 3.67 | 0.000 |
| lnFDI | −0.07175 | 0.02336 | −3.07 | 0.002 |
| Deve*lnEPU | −0.20816 | 0.041825 | −4.98 | 0.000 |
| Yunnan | 1.473401 | 0.277508 | 5.31 | 0.000 |
| Inner Mongolia | 2.619426 | 0.35334 | 7.41 | 0.000 |
| Beijing | −0.21481 | 0.101734 | −2.11 | 0.035 |
| Jilin | 0.647823 | 0.251551 | 2.58 | 0.010 |
| Sichuan | 2.240113 | 0.285604 | 7.84 | 0.000 |
| Tianjin | −0.40822 | 0.087541 | −4.66 | 0.000 |
| Ningxia | 0.071763 | 0.232847 | 0.31 | 0.758 |
| Anhui | 1.221871 | 0.174383 | 7.01 | 0.000 |
| Shandong | 3.967948 | 0.257148 | 15.43 | 0.000 |
| Shanxi | 2.01754 | 0.186097 | 10.84 | 0.000 |
| Guangdong | 3.958936 | 0.264439 | 14.97 | 0.000 |
| Guangxi | 1.227498 | 0.257793 | 4.76 | 0.000 |
| Xinjiang | 1.421676 | 0.389035 | 3.65 | 0.000 |
| Jiangsu | 3.504186 | 0.249844 | 14.03 | 0.000 |
| Jiangxi | 1.452037 | 0.246462 | 5.89 | 0.000 |
| Hebei | 2.532601 | 0.155007 | 16.34 | 0.000 |
| Henan | 2.1377 | 0.154104 | 13.87 | 0.000 |
| Zhejiang | 2.446262 | 0.258527 | 9.46 | 0.000 |
| Hainan | −0.32775 | 0.185984 | −1.76 | 0.078 |
| Hubei | 1.449128 | 0.22138 | 6.55 | 0.000 |
| Hunan | 1.678508 | 0.247464 | 6.78 | 0.000 |
| Gansu | 0.665623 | 0.338097 | 1.97 | 0.049 |
| Fujian | 2.142641 | 0.284653 | 7.53 | 0.000 |
| Guizhou | 1.337405 | 0.270246 | 4.95 | 0.000 |
| Liaoning | 2.115718 | 0.177887 | 11.89 | 0.000 |
| Chongqing | 0.795016 | 0.227269 | 3.5 | 0.000 |
| Shaanxi | 1.077528 | 0.206605 | 5.22 | 0.000 |
| Qinghai | 0.881767 | 0.472653 | 1.87 | 0.062 |
| Heilongjiang | 1.108825 | 0.286334 | 3.87 | 0.000 |

The investment effect of the EPU on environmental pollution emission is smaller than the consumption effect in all 30 provinces and autonomous regions of China, according to the estimation results of the regression model. The higher the EPU index, the lower the emission of environmental pollution. The coefficient of the interaction term is −0.208, which is significant at 1% significance level, indicating that in the developed provinces, the consumption utility brought by the EPU is greater than the investment effect brought by it. Moreover, the negative impact of the EPU index on environmental pollution emissions in developed provinces is greater than that in developing provinces. From the above point of view, Hypothesis 1 and Hypothesis 2 are verified.

## 4. Robustness Test

In order to test the correctness of hypothesis 1 and hypothesis 2 above, the comprehensive index of the environmental pollution emissions of each province, calculated by the entropy weight method, is used to replace the comprehensive index of the environmen-

tal pollution emissions of each province, calculated by the principal component analysis method for robustness test.

Similar to the previous panel regression of the EPU index, we control for the effects of night light data, total FDI actually utilized, and the interaction term. According to the F test and the Hausman test, the model adopts the fixed-effect model, and the model results are shown in the following Tables 6 and 7.

**Table 6.** Model selection.

|  | **Statistic** | *p*-**Value** |
|---|---|---|
| F test | 139.35 | 0.0000 |
| Hausman test | 16.43 | 0.0025 |

**Table 7.** Robustness test results.

| **Variables** | **Coefficient** | **Standard Error** | **T Statistic** | *p*-**Value** |
|---|---|---|---|---|
| C | 0.259815 | 0.054492 | 4.77 | 0.000 |
| lnEPU | −0.05430 | 0.011994 | −4.53 | 0.000 |
| lnVIIRS | 0.071134 | 0.017964 | 3.96 | 0.000 |
| lnFDI | −0.01259 | 0.00489 | −2.57 | 0.010 |
| Deve*lnEPU | −0.02066 | 0.008755 | −2.36 | 0.018 |
| Yunnan | 0.373003 | 0.05809 | 6.42 | 0.000 |
| Inner Mongolia | 0.637148 | 0.073963 | 8.61 | 0.000 |
| Beijing | −0.02496 | 0.021296 | −1.17 | 0.241 |
| Jilin | 0.179138 | 0.052656 | 3.40 | 0.001 |
| Sichuan | 0.487379 | 0.059784 | 8.15 | 0.000 |
| Tianjin | −0.05626 | 0.018325 | −3.07 | 0.002 |
| Ningxia | 0.070661 | 0.048741 | 1.45 | 0.147 |
| Anhui | 0.289955 | 0.036503 | 7.94 | 0.000 |
| Shandong | 0.692221 | 0.053828 | 12.86 | 0.000 |
| Shanxi | 0.524457 | 0.038955 | 13.46 | 0.000 |
| Guangdong | 0.600296 | 0.055354 | 10.84 | 0.000 |
| Guangxi | 0.283129 | 0.053963 | 5.25 | 0.000 |
| Xinjiang | 0.354054 | 0.081435 | 4.35 | 0.000 |
| Jiangsu | 0.556081 | 0.052299 | 10.63 | 0.000 |
| Jiangxi | 0.346526 | 0.051591 | 6.72 | 0.000 |
| Hebei | 0.615754 | 0.032447 | 18.98 | 0.000 |
| Henan | 0.441246 | 0.032258 | 13.68 | 0.000 |
| Zhejiang | 0.355041 | 0.054117 | 6.56 | 0.000 |
| Hainan | −0.02073 | 0.038931 | −0.53 | 0.594 |
| Hubei | 0.315809 | 0.046341 | 6.81 | 0.000 |
| Hunan | 0.35518 | 0.051801 | 6.86 | 0.000 |
| Gansu | 0.200169 | 0.070773 | 2.83 | 0.005 |
| Fujian | 0.336904 | 0.059585 | 5.65 | 0.000 |
| Guizhou | 0.320878 | 0.05657 | 5.67 | 0.000 |
| Liaoning | 0.505039 | 0.037236 | 13.56 | 0.000 |
| Chongqing | 0.185263 | 0.047573 | 3.89 | 0.000 |
| Shaanxi | 0.260138 | 0.043248 | 6.02 | 0.000 |
| Qinghai | 0.292821 | 0.098939 | 2.96 | 0.003 |
| Heilongjiang | 0.274615 | 0.059937 | 4.58 | 0.000 |

It can be seen from the robustness test that the negative impact of the EPU index on the environmental pollution emission composite index is still significant. The coefficient of the EPU index is −0.054, which is significant at the significance level of 1%. That is, the higher the EPU index, the lower the composite index of environmental pollution emissions, and the lower the environmental pollution emissions. Similarly, the interaction term Deve*lnEPU coefficient is −0.021 at the significance level of 5%. By studying the difference between economically developed provinces and those with an average economic development

level, it can be seen that the negative impact of the EPU index on environmental pollution emissions is 1.38 times that of those with an average economic development.

The conclusions of the robustness study in this paper are consistent with those of the previous empirical study. Therefore, it can be shown that the regression results of this paper are robust and effective.

## 5. Discussion and Conclusions

### 5.1. Discussion

In regard to the relationship between EPU and environmental pollution, the academic circle has not reached a unified conclusion. Scholars have considered the interaction between EPU and environmental pollution from different perspectives. Some scholars believe that there is a positive correlation between carbon dioxide emissions and EPU (Wang et al. [2], Jiang et al. [3]). Some scholars believe that there is a negative correlation between carbon dioxide and EPU (Chen et al. [5], Danish et al. [6]). Other scholars argue that the relationship between carbon dioxide and EPU is different in the short term and the long term (Adedoyin et al. [7]). The reasons for the differences in the above conclusions are presumed to be that there are some exogenous variables that affect the results, such as resources and economic development level. Different researchers have controlled for different economic variables according to different research purposes, or have adopted different measurement methods, leading to different results.

In terms of the research on the ways in which EPU affects environmental pollution, most scholars' views are consistent with the research of Wang et al. [2]; that is, the influence is generated through the "investment effect" and the "consumption effect".

However, the existing literature rarely examines the heterogeneity that exists within an economic entity. On the basis of the existing literature, and in an attempt to fill the gaps in the existing literature, we not only studied the causal effect of EPU on environmental pollution, but also investigated whether the differences of economic development levels, measured by nighttime light data, have an effect on the impact results.

The research of this paper has a positive significance for China—a country with a huge economic volume and a complex economic development level and structure—in realizing its long-term goal of "double carbon". Similarly, the research of this paper is worthy of reference for other countries with similar development levels as China. However, this paper was not broad enough to study the sustainable development of the green economy and the realization of the "dual carbon" goal in the future under the fluctuation mode of different scenarios of economic policy uncertainty. Future scholars can conduct research from the above perspective. Economic policy uncertainty, of course, creates different goals for the future situation of the model. For example, with what and how to measure the time of the target is a problem worthy of studying. Because of the unclear circumstances of the world economic situation, the "double carbon" goal has a very vital significance for China, India, and other countries.

### 5.2. Conclusions

The comprehensive pollution emission index of each province is taken as the explained variable, and the EPU of China is taken as the core explanatory variable. By constructing panel data of the 30 provinces in mainland China from 2008 to 2020, the influence of the EPU on the comprehensive pollution emission index of each province is analyzed and studied. In order to highlight the heterogeneity among provinces with different levels of economic development, nighttime light data, total amount of FDI actually utilized, and the interaction terms were taken as control variables.

It can be seen from the analysis of the empirical results in this paper that, within China, the economic policy uncertainty index shows a reverse relationship with the comprehensive index of the environmental pollution emissions of provinces and autonomous regions, which is consistent with the results obtained by Danish et al. [6] and Chen et al. [5]. At the same time, the literature part of this paper shows that the investment effect can increase

the comprehensive environmental pollution emission index, while the consumption effect can reduce the comprehensive environmental pollution emission index. Therefore, the establishment of hypothesis 1 in this paper indicates that the investment effect brought by the increase in economic policy uncertainty is smaller than the consumption effect. This is consistent with the conclusions of Wang et al. [2]. From the perspective of the investment effect and the consumption effect caused by economic policy uncertainty, this study is a further extension of the research of Adams et al. [4] and Chen et al. [5].

The establishment of hypothesis 2 in this paper is completely different from the study of Chen et al. [5]. Chen et al. [5], through the study of 15 countries' data, found that economic policy uncertainty among emerging market countries, such as China, India, and Brazil, had more of an impact on the environmental pollution emissions than in economically developed countries, such as America, Germany, and Japan, whose influence on environmental pollution is bigger, with the former being about three times the latter. However, the research results of this paper show that within regional economies of China, the impact of the EPU on environmental pollution emissions in the economically developed provinces is 2.98 times greater than that in the provinces with an average economic development level. In addition, this paper adopts the comprehensive pollution emission index of each province, calculated by the entropy weight method, to conduct the robustness test, and the results are also significant and effective.

**Author Contributions:** Conceptualization, W.Y.; methodology, Y.Z.; validation, Y.Z.; formal analysis, W.Y.; investigation, Y.H.; resources, Y.Z.; writing—original draft preparation, Y.Z.; writing—review and editing, W.Y. All authors have read and agreed to the published version of the manuscript.

**Funding:** Supported by the China Agriculture Research System of MOF and MARA (CARS-47).

**Institutional Review Board Statement:** Not applicable.

**Informed Consent Statement:** Not applicable.

**Data Availability Statement:** Data available in a publicly accessible repository that does not issue DOIs. Publicly available data sets were analyzed in this study. These data can be found here: [http://www.ppmandata.cn/], accessed on 11 March 2022.

**Conflicts of Interest:** The authors declare no conflict of interest.

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
