# Peer review of "Heterogeneous Impact of Economic Policy Uncertainty on Provincial Environmental Pollution Emissions in China"

_sustainability, doi:10.3390/su14094923_

Round 1

Reviewer 1 Report

The research questions raised by the author are worthy of recognition and conform to the hotspots. But there are some problems:

  1. There are many detailed errors in the text, such as no space after commas, inconsistent writing of fixed vocabulary (eg: P-value), lack of full spelling explanations for abbreviations, etc.
  2. Lines 38-42 are duplicated with line 36, it is recommended to delete them.
  3. Lines 60-62: The three main point sentences can be expressed more clearly.
  4. Line 67: It is recommended to explain the relationship between WUI and EPU.
  5. Lines 81-122: The context of these two paragraphs needs to be explained. What is the point of analyzing them?
  6. Lines 182-183: The meaning of comprehensive index of environmental pollution emission is not explained.
  7. It is recommended to separate the hypothesis part from the introduction and make it a separate chapter.
  8. Lines 194-195: Delete “so”.
  9. Lines 205-207: Supplement the full spelling of DMSP and VIIRS.
  10. Lines 215-217: The full spelling of CE appears twice.
  11. Lines 240-241: There should be no Chinese in the picture, and the pictures are too small to see clearly.
  12. Lines 253 & 262: Coefficient letters should be different for the two models.
  13. Line 298: The results section lacks an explanation of the investment effects and consumption effects mentioned above.
  14. Line 346: You should write the discussion first and then the conclusion. The analysis of the investment effects and consumption effects mentioned above is missing from the discussion.

Author Response

Thank you for your valuable suggestions for revision. I have revised them point by point. Please see the attachment.

Reviewer 2 Report

Dear authors,

The text needs an intensive review. It seems that the text has been written very quickly and carelessly.

In many sentences the space after point and/or comma is missing. It is not punctual; I have seen this error repeated in all the text. Sometimes, the space between words is also missing.

The references inserted in the text are not correctly cited. According to the “Instruction for authors”: “In the text, reference numbers should be placed in square brackets [ ], and placed before the punctuation; for example [1], [1–3] or [1,3]. For embedded citations in the text with pagination, use both parentheses and brackets to indicate the reference number and page numbers; for example [5] (p. 10). or [6] (pp. 101–105).” Please check all references.

Contractions are not correct in formal/scientific writing, e.g: “China's economic development” that has to be changed to “the economic development of China”. Please, check all the text.

The DISCUCCION must be always before than CONCLUSIONS.

I recommend that a reviewer with more experience in maths and economic policy revise the analysis and the text because I am not specialist in this field. 

Other comments or corrections are highlighted and commented in the attached PDF.

Author Response

(The authors gave the same response as above.)

Round 2

Reviewer 1 Report

The current Discussion is not comprehensive enough, and the Conclusion is not concise enough and covers what should be written in the Discussion section. It is recommended that the authors further investigate what should be written in each of these two sections.

Author Response

Dear Professor:

Thank you very much for your valuable suggestions. I have put forward modifications to your suggestions.If there are any comments, I will continue to revise. Thank you very much.

Reviewer 2 Report

I consider that the manuscript has been sufficiently improved to warrant publication in Sustainability.

Thank you

Author Response

Thank you very much for your advice. Can you select "I would like to sign my review report"? Whether there are other problems, I will continue to modify